# Differential Stress Responses to Rice Blast Fungal Infection Associated with the Vegetative Growth Phase in Rice

**DOI:** 10.3390/plants14020241

**Published:** 2025-01-16

**Authors:** Takuma Koyama, Takumi Tezuka, Atsushi J. Nagano, Jiro Murakami, Takanori Yoshikawa

**Affiliations:** 1Department of Agriculture, Kyoto University, Kyoto 606-8502, Japan; 2Department of Genomics and Evolutionary Biology, National Institute of Genetics, Mishima 411-8540, Japan; 3Faculty of Agriculture, Ryukoku University, Otsu 520-2194, Japan; 4Institute for Advanced Biosciences, Keio University, Tsuruoka 997-0035, Japan; 5Faculty of Agriculture, Kibi International University, Minami-Awaji 656-0484, Japan; 6Graduate School of Agriculture, Kyoto University, Kyoto 606-8502, Japan

**Keywords:** rice, vegetative phase transition, rice blast, transcriptome, weighted gene coexpression network analysis (WGCNA)

## Abstract

During vegetative growth, plants undergo various morphological and physiological changes in the transition from the juvenile phase to the adult phase. In terms of stress resistance, it has been suggested that plants gain or reinforce disease resistance during the process of maturation, which is recognized as adult plant resistance or age-related resistance. While much knowledge has been obtained about changes in disease resistance as growth stages progress, knowledge about changes in plant responses to pathogens with progressing age in plants is limited. In this study, we experimentally compared rice blast resistance in rice leaves sampled from plants at different growth phases. The results indicate differential infection progression and fungal status depending on growth stage. Transcriptome analysis following blast fungus infection revealed that several genes involved in the defense response were upregulated in both the juvenile and intermediate stage, but the expression changes of many genes were growth phase-specific. These findings highlight differences in rice leaf stress responses to blast infection at different growth stages.

## 1. Introduction

The plant life cycle can be broadly divided into embryogenesis, vegetative, and reproductive phases. The vegetative phase is characterized by the transition from juvenile to adult plants, which involves diverse morphological and physiological changes. For example, during the vegetative phase transition in *Arabidopsis thaliana*, the petiole becomes shorter, the leaf shape changes from round to oval, the distribution of trichomes on the leaf surface changes, leaf cell numbers increase, and leaf cells decrease in size [1]. In the case of rice, the size of the leaves increases, the midrib develops, and the leaf shape becomes slender as the growth stage progresses [2]. In addition, leaf photosynthetic activity increases and the trophism switches from heterotrophic to autotrophic [2,3]. From the morphological and physiological changes, it is considered that the first and second leaf stages are the juvenile phase, the third to fifth leaf stages are intermediate, and the later stage is the adult phase in rice [2].

Various endogenous and exogenous signals regulate the vegetative phase transition, such as day length, temperature, and gibberellin and sugar contents [4,5]. Of these signals, *microRNA156* (*miR156*) plays a key role in the molecular genetic mechanisms underlying the vegetative phase transition. miRNA consists of 20–24 nucleotides of small non-coding RNAs that regulate the expression of target genes with complementary sequences through mRNA cleavage and transcriptional repression. A few miRNAs are conserved among diverse plant species [6,7], and gene regulation by these RNAs is indispensable for diverse biological processes. *miR156* regulates plant morphological and physiological processes such as flowering, plastochron determination, inflorescence structural formation, and branching through the regulation of genes in the *SQUAMOSA PROMOTER-BINDING PROTEIN-LIKE* (*SPL*) family [8,9,10]. SPLs are plant-specific transcription factors that bind to DNA via the *SQUA* promoter-binding protein domain to regulate gene expression [11,12]. In total, 19 *SPLs* have been identified in rice, of which 11 are targets of *miR156*. As plants progress from the juvenile to the adult phase, *miR156* expression levels gradually decrease, whereas those of *miR172*—another phase change regulator indirectly regulated by *miR156*—increase. This temporal change in *miR156* and *miR172* expression levels induces an *SPL* increase and a decrease in the transcription levels of *AP2*, a group of transcription factors targeted by *miR172*, resulting in the emergence of the mature phenotype [13,14,15,16,17].

Generally, plants are more susceptible to disease in the early phase than in the late phase, and resistance to disease increases during the process of maturation [18,19,20]. This increase could be due to the gain or reinforce the ability to control the infection and/or proliferation of pathogen, which is known as adult plant resistance or age-related resistance [21,22,23]. The resistance of rice to bacterial blight or the blast fungus usually increases with plant and leaf age [18,19,20]. The first genetic evidence linking *miR156* to age-related resistance was obtained from the maize mutant *Corngrass1* (*Cg1*) [24]. In *Cg1*, resistance to common rust and European corn borer was delayed coordinately with the delay of adult phase characteristics due to the overexpression of *miR156* [24,25]. Previous studies have demonstrated that *miR156* is involved in both biotic and abiotic stress tolerance. For example, *miR156* overexpression in *A. thaliana* enhances salt and drought stress [26]. These effects are caused by an increase in anthocyanin biosynthesis induced by the regulation of the anthocyanin biosynthesis gene *DFR* via the *miR156-SPL9-DFR* module. *miR156* has also been reported to confer heat stress tolerance in alfalfa through *SPL13* downregulation [27]. In rice, the overexpression of *miR156* increased rice blast susceptibility, whereas the expression of a target mimic of *miR156* enhanced resistance to rice blast fungus [28]. This alteration in resistance is attributable to changes in the expression levels of *WRKY45*, a WRKY transcription factor that regulates many resistance genes, downstream of *SPL14*.

Although much knowledge has been obtained regarding the changes in disease resistance through the maturation of plants, there is limited understanding of changes in plant responses to pathogens with progressing plant age. In this study, we treated rice leaves at different growth stages with the hemibiotrophic fungus *Pyricularia oryzae*, which causes severe rice blast disease worldwide, to compare rice leaf stress responses between the growth stages. The results indicate differential infection progression and fungal status depending on growth stage, and the transcriptome analysis of infected leaves revealed distinct gene expression profiles in juvenile and older leaves. These findings highlight differences in rice leaf stress responses to *P. oryzae* infection between the growth stages.

## 2. Results

### 2.1. Evaluation of Rice Blast Resistance in Leaves at Different Growth Stages

To assess rice blast resistance in rice leaves from plants at different growth phases, we quantified the area of the lesions caused by rice blast fungus and estimated the fungal biomass based on the amount of blast *MHP1* gene DNA in leaves [29]. The second, third, fourth, and fifth leaves were sampled from plants and inoculated with conidia. Based on measurements performed 6 days later, the average lesion area was found to decrease from the juvenile to the intermediate stages (Figure 1A). Significant differences were detected between the second and fourth, second and fifth, and third and fifth leaves (Figure 1B). Our lesion area comparisons imply that juvenile leaves tend to show more severe disease symptoms than the older leaves.

Next, the fungal biomass on each leaf was estimated using the real-time polymerase chain reaction (qPCR). Considering that lesions on juvenile leaves were larger than those on later leaves, we expected that fungal growth would be repressed on adult leaves. However, the second and third leaves contained significantly less rice blast fungal biomass than the fourth and fifth leaves (Figure 1C). Although the estimated relative fungal biomass basically showed a positive correlation with the lesion area, the relationship between the two in the fourth and fifth leaves was clearly different from that in the second and third leaves (Figure 1D), and this difference suggested that older leaves contained more fungal biomass per lesion area. To investigate the cause, the lesion areas of the second and fifth leaves were stained with trypan blue and observed under an optical microscope. Compared with the second leaf, the hyphae with conidia were more densely spread around the inoculated spot in the fifth leaf (Figure 1E,F). Therefore, it was assumed that the difference in hyphal density was the cause of the discrepancy between the lesion area and fungal biomass. These results suggest that the status of the infecting fungus differs depending on the leaf age and imply that plants at different stages exhibit different responses to rice blast infection. Thus, juvenile leaves appear more susceptible to damage by blast infection than adult leaves, as supported by the presence of larger lesion areas on the second leaves than on the fifth leaves (Figure 1A,B), and blast fungal biomass increases faster on adult leaves than on juvenile leaves, as supported by the higher hyphal density on fifth leaves than on second leaves (Figure 1E,F). We hypothesize that these differences are likely the result of complex physiological differences between juvenile and adult leaves, such as stored chemical compounds or transcriptome alterations in response to fungus infection.

### 2.2. Transcriptome Dynamics in Rice Leaves at Different Stages Under Rice Blast Fungus Infection

To compare gene expression responses to rice blast infection between different growth phases, we conducted the transcriptome analysis of inoculated and control leaves. Differentially expressed genes (DEGs) were identified by comparing inoculated second leaves with control second leaves and by comparing inoculated fourth leaves with control fourth leaves based on the criteria of adjusted *p* < 0.05 and log_2_ fold change > 1. We identified 123 DEGs in second leaves (65 upregulated; 58 downregulated), and 417 DEGs in fourth leaves (228 upregulated; 189 downregulated), with 12 DEGs shared by both second and fourth leaves (Table 1; Figure 2A).

Among the 12 DEGs shared by second and fourth leaves under blast infection, 10 were upregulated in both second and fourth leaves, including *Os03g0661600*, *Os04g0688100*, *Os08g0135900*, *Os10g0416500*, and *Os12g0437800*, which are thought to be involved in plant defense responses against rice blast disease. *Os03g0661600* resembles alpha-amylase/trypsin inhibitors, which inhibit fungal pathogen growth. *Os04g0688100* encodes peroxidase, which is induced by rice blast infection [30]. *Os08g0135900* is similar to a tryptophan synthase beta chain 1 gene. Tryptophan is an amino acid that acts as a precursor of substances such as auxin, melatonin, and serotonin; tryptophan synthesis activation has been reported to be involved in rice defense responses against the pathogenic fungus *Bipolaris oryzae* [31,32]. *Os10g0416500* resembles a chitinase 1 precursor gene. Chitinase is a pathogenesis-related protein that degrades chitin, a primary component of the fungal cell wall, thereby defending plants from pathogens. *Os12g0437800* is similar to a maize proteinase inhibitor gene that encodes a pathogenesis-related protein in maize; its expression is induced by pathogen invasion [33]. As these genes were upregulated in both the second and fourth leaves, we expected that rice defense responses against blast disease infection would be induced similarly in the inoculated second and fourth leaves. However, normalized counts of *Os12g0583300* were zero in control leaves, whereas gene expression was strongly induced in inoculated leaves (Table 1). *Os12g0583300* encodes peptidase aspartic, a catalytic domain-containing protein. A previous study suggested that extracellular aspartic protease was involved in disease resistance signaling in *Arabidopsis*, and many studies have investigated the role of aspartic protease in plant defense responses [34,35]. *Os12g0583300* upregulation in the inoculated second and fourth leaves implies that aspartic protease is involved in rice defense responses against blast disease. Among the twelve DEGs shared by the second and fourth leaves under blast infection, only one gene, *Os07g0218200*, was downregulated in second leaves and upregulated in fourth leaves. *Os07g0218200* is similar to the terpene synthase 7 gene. Terpene is a secondary metabolite that acts as an antimicrobial compound during pathogen infection [36]. This differential gene expression change implies that during infection, certain defense reaction pathways are suppressed in the second leaves but induced in the fourth leaves.

Next, we conducted a Gene Ontology (GO) enrichment analysis of the 111 DEGs identified only in second leaves (Appendix A), with separate analyses performed for the 55 upregulated and 56 downregulated genes. The upregulated genes were significantly enriched in stress, stimulus, and defense responses, as well as in cellular catabolic processes (*p* < 0.05; Figure 2B). We did not detect the significant enrichment of the downregulated genes.

Next, we conducted the GO enrichment analysis of the 217 upregulated and 188 downregulated DEGs identified only in fourth leaves (Appendix A). The upregulated genes were significantly enriched in oxidation–reduction processes and chemical and defense responses (Figure 2C), as well as processes involving secondary metabolites such as isoprenoid, diterpenoids, terpenoids, and serotonin. The downregulated genes were significantly enriched in photosynthesis-related processes such as photosynthesis, carbon fixation, photosynthetic electron transport chain function, and photosystem II stabilization (Figure 2D), indicating the induction or suppression of pathways involving secondary metabolites and photosynthesis in fourth leaves by rice blast infection.

Because we detected twice as many DEGs in fourth leaves as in second leaves (Figure 2A), we hypothesized that these groups of genes form networks that coordinate gene up- and downregulation in response to rice blast infection. To test this hypothesis, we performed weighted gene co-expression network analysis (WGCNA) of specific DEGs detected in fourth leaves to identify gene co-expression networks. A hierarchical clustering tree identified four distinct modules: blue, yellow, brown, and turquoise (Figure 3A). The GO analysis of genes belonging to the brown and yellow modules showed no GO enrichment; however, pathways related to secondary metabolites and photosynthesis were enriched in the blue and turquoise modules, respectively (Appendix A).

Therefore, we concluded that these modules contained functionally related networks and subjected the genes of these modules to further analysis.

All genes included in the blue module were upregulated by rice blast infection in the fourth leaves (Appendix A). Furthermore, many of the genes associated with secondary-metabolism-related GO terms enriched in the 217 upregulated DEGs that were specific to fourth leaves belonged to the blue module (Appendix A). Therefore, we concluded that genes related to secondary metabolism and amino acid synthesis pathways were concentrated in the blue module. To identify a network with higher co-expression among the blue modules, we extracted only those edges with weight > 0.081 and constructed a network consisting of 57 genes (Figure 3B). Genes within this network that were co-expressed with many other genes were identified as hub genes, among which *Os12g0268000*, *Os06g0549900*, *Os10g0346300*, *Os07g0182100*, and *Os03g0570100* were the most prominent. These genes encode Cytochrome P450 71P1(CYP71P1), FAD-linked oxidase, Laccase 17, Tryptophan synthase alpha subunit, and Cytochrome P450 79A1, respectively, and are involved in secondary metabolite biosynthetic pathways.

All genes within the turquoise module were downregulated by rice blast infection in fourth leaves (Appendix A). Furthermore, many genes associated with photosynthesis-related GO terms enriched in the 188 downregulated DEGs specific to fourth leaves were included in the turquoise module (Appendix A). Thus, genes related to photosynthesis and carbon fixation pathways appeared to be concentrated in the turquoise module. Therefore, we extracted edges with weights > 0.141 and constructed a network consisting of 53 genes (Figure 3C), among which *Os12g0274700*, *Os12g0291100*, *Os01g0200700*, and *Os01g0200700* appeared to be major hub genes. *Os12g0274700* and *Os12g0291100* encode ribulose-1,5-bisphosphate carboxylase/oxygenase (RubisCO). *Os01g0200700* and *Os03g0278900* encode metallothionein I-3A and ATPase, respectively. Metallothionein is involved in stress responses such as protection against reactive oxygen species (ROS) and metals [37]. *Os03g0278900* was found to be downregulated together with *Os12g0274700* under chilling stress [38].

These results imply that rice exhibits differential responses to blast infection associated with the vegetative growth phase. Transcriptome analysis revealed gene expression changes that were common to juvenile and older leaves, but the expression levels of secondary metabolites and photosynthesis-related networks were significantly altered in fourth leaves, which appears to explain the difference in numbers of DEGs between different growth phases in rice.

## 3. Discussion

We infected rice leaves sampled from plants at different growth phases with rice blast conidia and compared lesion expansion and fungal biomass in leaves (Figure 1). The lesions tended to be larger on leaves of juvenile plants (Figure 1A,B), whereas fungal biomass was higher on those from older plants (Figure 1C). This discrepancy can be explained by the more fungal biomass per lesion area in older plants than in juvenile plants due to the higher hyphal density (Figure 1D–F). Thus, rice leaves appeared to respond differently to blast infection depending on the vegetative growth phase. Transcriptome analysis showed differences in gene expression patterns between infected second and fourth leaves. Following rice blast infection, the expression of genes involved in the plant defense response to rice blast fungus was upregulated specifically in second leaves. In contrast, secondary metabolism-related genes were upregulated (Figure 2C) and photosynthesis-related genes were downregulated (Figure 2D) specifically in fourth leaves. These results imply that leaves from juvenile and older plants exhibit different gene expression patterns in response to rice blast infection. Therefore, we conclude that plant defense mechanisms against rice blast infection may be altered by the vegetative phase transition.

To date, several compounds have been reported to accumulate during the juvenile phase of vegetative growth. For example, DIMBOA is a strong antibacterial compound in maize; this benzoxazinoid is present in high concentrations after germination, but content decreases with plant age [39,40]. Antibacterial compounds such as DIMBOA may protect juvenile rice plants from blast infection. Future studies should test this hypothesis through metabolomic analyses, which can comprehensively compare accumulated compounds between different growth phases.

In the fourth leaves, we detected the upregulation of genes involved in the secondary metabolism and downregulation of genes associated with photosynthesis (Figure 2). WGCNA identified *CYP71P1* as a hub gene within the blue module (Figure 3B). Rice *cyp71p1* mutants exhibit a phenotype resembling rice blast lesions. *CYP71P1* is induced by chitin elicitors that trigger a defense response in rice and catalyze the serotonin biosynthesis pathway [41]. A previous study found that serotonin accumulated at the site of *P. oryzae* infection in rice, but at lower levels in *cyp71p1* mutants than in wild-type plants [31]. Serotonin accumulation is thought to confer blast fungus resistance by strengthening cell walls, inducing cell death, and promoting the expression of resistance genes [31,41]. Therefore, serotonin may enhance plant resistance to rice blast fungus by regulating downstream genes in the fourth leaves. The blue module may encompass genes involved in the serotonin-mediated pathway for blast resistance.

In this study, WGCNA identified two RubisCO genes as hub genes within the turquoise module (Figure 3C). Previous studies have reported that the expression of photosynthesis-related genes is downregulated under biotic stress, which induces ROS accumulation [42,43]. Plants under stress exhibit a hypersensitive response to ROS accumulation as a strategy to prevent the spread of infection, such that ROS accumulation is interpreted as being a signal of plant defense responses. The enrichment of downregulated DEGs in photosynthesis-related pathways in fourth leaves may indicate the induction of defense responses involving ROS.

This study is the first to demonstrate that rice exhibits different strategies against blast infection depending on the vegetative growth phase. Future studies comparing leaf metabolites between growth stages and defense responses across diverse genetic backgrounds will provide deeper insights into the mechanisms underlying phase-specific blast resistance in rice.

## 4. Materials and Methods

### 4.1. Rice Growth Conditions

The *japonica* rice (*Oryza sativa* L. ssp. japonica) cultivar ‘Nipponbare’ was used in this study. Rice seeds were sterilized in hot water at 60 °C for 10 min, followed by soaking in triflumizole solution for 1 day. Sterilized seeds were then soaked in water for 2 days for germination, planted in soil in a cell tray, and then placed in a growth chamber (Nippon Medical & Chemical Instruments, Osaka, Japan) at 28 °C under a 14 h light/10 h dark light cycle. The definition of ‘juvenile phase’ and ‘adult phase’ followed Itoh et al. (2005) [2]; the first and second leaf stages are the juvenile phase, the third to fifth leaf stages are intermediate, and the later stage is the adult phase. Under our growth conditions, leaves turned yellow when the rice plants reached the sixth leaf stage due to the long duration (three weeks) in the growth chamber; therefore, rice plants at the second to fifth leaf stages were used. The latest fully developed leaves at each leaf stage from separate individuals were used for the experiments. Leaf stages were determined based on the emergence of the tip of the next leaf from the leaf sheath. For example, when the tip of a third leaf appeared, the plant was determined to be in the second leaf stage.

### 4.2. *Pyricularia oryzae* Cultivation and Conidium Formation

*Pyricularia oryzae* strain Hoku1 (Japanese race no. 007) was used in this study, as it is compatible with the ‘Nipponbare’ rice cultivar. The cultivation protocol for *P. oryzae* followed that of Murakami et al. [44]. Blast fungus was cultivated on potato dextrose agar medium at 23 °C. To obtain conidia, fungal hyphae were transferred to oatmeal medium containing 10 g oatmeal powder, 10 g agar, 2.5 g sucrose, and 500 mL water and maintained at 23 °C for 1 week. Aerial hyphae were then removed by pouring sterilized water over the medium and gently scratching the surface with a sterilized microtube (Appendix A). The water was then removed from the surface, and the oatmeal medium was placed under ultraviolet A light at 23 °C for 4 days for conidia formation.

### 4.3. Inoculation with Rice Blast Fungus

Conidia formation was induced as described in the previous section. The conidia were counted using a hemocytometer (Appendix A), and their concentrations were adjusted to 5 × 10^5^ conidia/mL with sterilized water containing 0.01% Tween20. A corresponding control experiment was conducted using only sterilized water containing 0.01% Tween20.

Plant cell walls thicken with plant growth, forming a physical defense against pathogen invasion. Although spray inoculation is a common method of evaluating rice blast disease resistance, it can be affected by cell wall thickness. Therefore, we conducted punch wound inoculation to evaluate physiological differences accurately when comparing rice blast disease resistance between groups. Each leaf blade sample was wounded (diameter, 1.6 mm) using a crafting punch that was modified to crush cells without punching a hole through the leaf. We applied 3 μL of conidial suspension to each wound. The inoculated plants were placed in boxes in the dark for 1 day at 23–25 °C at 100% humidity; the light conditions were then changed to a 14 h light/10 h dark cycle. Disease severity was measured 6 days after inoculation. Images of this process are presented in Appendix A.

### 4.4. Disease Severity Assessment by Lesion Area

Lesions were photographed using a digital camera 6 days after inoculation (e.g., Appendix A). The areas of lesions (brown or yellow spots) were measured using ImageJ software (ver. 1.5.3) (National Institutes of Health, Bethesda, MD, USA). A few samples were excluded from analysis based on Smirnov–Grubbs tests [45] of the lesion area.

### 4.5. Disease Severity Assessment by qPCR

The entire leaf blades, including all lesions, were sampled 6 days after inoculation; the samples were stored in a freezer. Leaf samples were frozen with liquid nitrogen and homogenized with cell disruptor beads (Yasui Kikai, Osaka, Japan). We added 500 μL of TPS buffer containing 100 mM Tris-HCl (pH 8.0), 1 M KCl, and 10 mM ethylenediaminetetraacetic acid to the sample tubes, and stored them at 70 °C for 30 min, followed by centrifuging at 15,000 rpm for 10 min at 4 °C. Then, 320 μL of supernatant was obtained from each sample, to which 320 μL of isopropyl alcohol was added, and the samples were inverted and centrifuged at 15,000 rpm for 10 min at 4 °C. The precipitated DNA was rinsed with 500 μL of 70% ethanol, dissolved with sterilized distilled water, and subjected to qPCR. To quantify the fungal biomass in each sample, qPCR was conducted using KOD SYBR qPCR MIX (Toyobo, Osaka, Japan) on a LightCycler 480 System (Roche, Basel, Switzerland). Primers (forward: 5′-GTCCTGCCCATCGTAAGTTC-3′, reverse: 5′-TTGCAGAGGATACCCTGGTC-3′) were used to amplify 113 bps of the *MHP1* gene of *P. oryzae*. *MHP1* is a single-copy gene that encodes hydrophobin, which is essential for host invasion [29]. A standard curve was created through the serial dilution of DNA extracted from the second leaf (Appendix A). The relative *MHP1* gene DNA concentration of each sample to that of the second leaf was determined based on this standard curve and calculated crossing point values. Three technical replicates were analyzed for each experimental group.

### 4.6. Trypan Blue Staining

Leaf blades were sampled 6 days after inoculation for microscopy observations. For decolorization, the leaf samples were soaked in alcoholic lactophenol, composed of 10 mL lactophenol and 20 mL of ethanol, and then rinsed with a lactophenol solution containing 10 mL phenol, 10 mL glycerol, 10 mL lactic acid, and 10 mL water. Then, leaf samples were added to a trypan blue solution (250 μg/mL trypan blue in lactophenol), heated at 100 °C for 2 min, and cooled at room temperature for 1 h [46]. Stained leaf blades were soaked in lactophenol for decolorization for 1 h and mounted in glycerol solution. Lesion areas were observed under an optical microscope.

### 4.7. Inoculation and Sampling of P. oryzae for RNA Sequencing (RNA-Seq) Analysis

We used *P. oryzae* strain Hoku1 in this experiment. A conidial suspension (5 × 10^5^ conidia/mL) containing 0.01% Tween20 was prepared, and a corresponding control experiment was conducted using sterilized water containing 0.01% Tween20. Three to sixteen wounds were made on each leaf blade using the modified punch, and each wound was treated with 3 μL of conidial suspension (Appendix A). Inoculated plants were placed in humid boxes at 23–25 °C in the dark for 24 h, and then under a 14 h light/10 h dark cycle for the following 24 h. At 48 h after inoculation, leaf discs with a diameter of 2 mm around the wound were sampled using a hole punch (Appendix A) and immediately frozen with liquid nitrogen. Samples were stored at −80 °C until RNA extraction.

### 4.8. RNA-Seq Analysis

Leaf samples were frozen with liquid nitrogen and homogenized using a multi-bead shocker (Yasui Kikai). Then, total RNA was extracted with TRIzol Reagent (Invitrogen, Carlsbad, CA, USA) following the manufacturer’s instructions. RNA samples were sequenced with low-cost, easy sequencing (Lasy-seq) using the Illumina platform [47]. Three biological replicates were analyzed for each experimental group.

Quality checks of the RNA-seq reads were conducted using Fastp v0.23.2. The preprocessed reads were mapped against the reference ‘Nipponbare’ genome sequence (IRGSP-1.0_genome.fasta) using HISAT2 v2.2.1. The number of reads mapped on each gene was determined based on gene annotation data downloaded from the RAD-BP vIRGSP-1.0 2021-11-11 using featureCounts subread v2.0.1.

### 4.9. Identification of DEGs

Gene expression changes were calculated using DESeq2 v1.40.2 by comparing the counts of each gene between control and inoculated samples from second and fourth leaves, respectively. Count normalization was conducted using the default settings. DEGs were identified based on significance thresholds of adjusted *p* < 0.05 and log_2_(fold change) > 1.

### 4.10. GO Enrichment Analysis

GO enrichment analysis was conducted using the ShinyGO web application (v0.741) [48]. Upregulated and downregulated DEGs for the second and fourth leaves were identified by ShinyGO application, at a significance threshold of *p* < 0.05. Only the top 30 GO terms are reported in this study.

### 4.11. WGCNA

WGCNA was conducted using the R package WGCNA v1.72.5, based on normalized read counts for the 405 DEGs specifically identified in fourth leaves. A topological overlap matrix (TOM) was calculated using an optimal soft threshold determined by the “pickSoftThreshold” function. Hierarchical clustering was performed using the dissimilarity TOM, which was obtained by subtracting the TOM from 1. Modules were detected from the obtained dendrogram using the “cutreeDynamic” function with the option deepSplit = 4, and the co-expression networks of genes included in the modules were constructed using Cytoscape v3.10.3.

## Figures and Tables

**Figure 1 plants-14-00241-f001:**
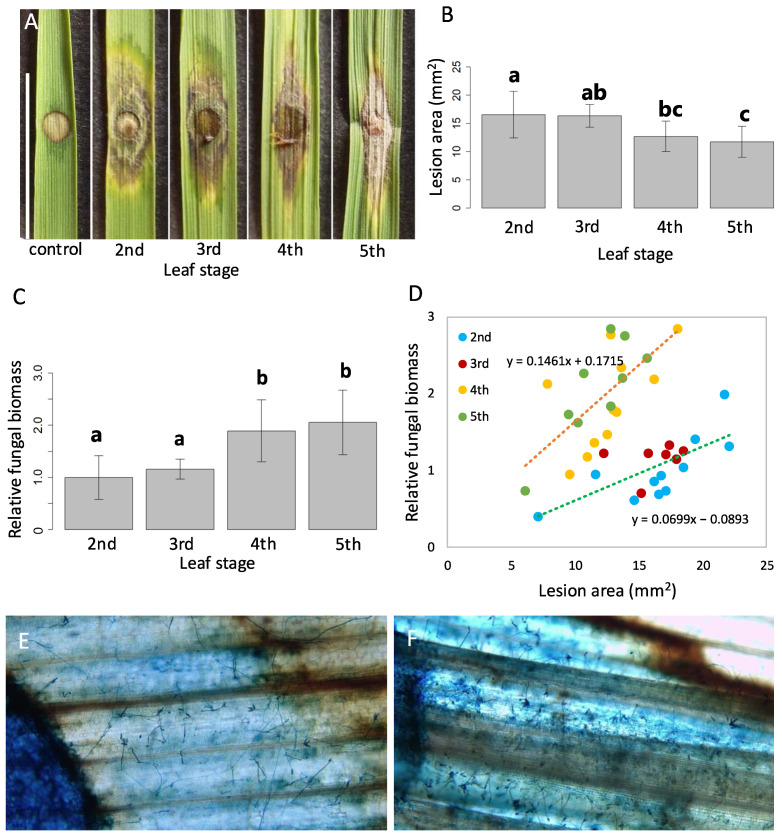
*P. oryzae* infection in different growth stages of rice. (**A**) Lesions on leaves at different plant growth stages. The control leaves were treated with sterilized water. Scale bar = 10 mm. (**B**) Average lesion areas for leaves at different stages. (**C**) Estimated relative fungal biomass on leaves sampled from plants at different stages. (**D**) Relationship between lesion area and estimated relative fungal biomass at different plant growth stages. The green dashed line indicates the regression line for the mixed group of second and third leaves, and the orange for the mixed group of fourth and fifth leaves. (**E**,**F**) Trypan blue staining of lesions on second leaf (**E**) and fifth leaf (**F**) (**E**,**F**; 100×). The dark blue regions on the left are the wounds for the fungus infection. Error bars in (**B**,**C**) indicate standard deviation; different letters above bars indicate significant differences (*p* < 0.05; Tukey–Kramer test). Sample sizes were as follows: second leaves, *n* = 11; third leaves, *n* = 7; fourth leaves, *n* = 11; and fifth leaves, *n* = 9.

**Figure 2 plants-14-00241-f002:**
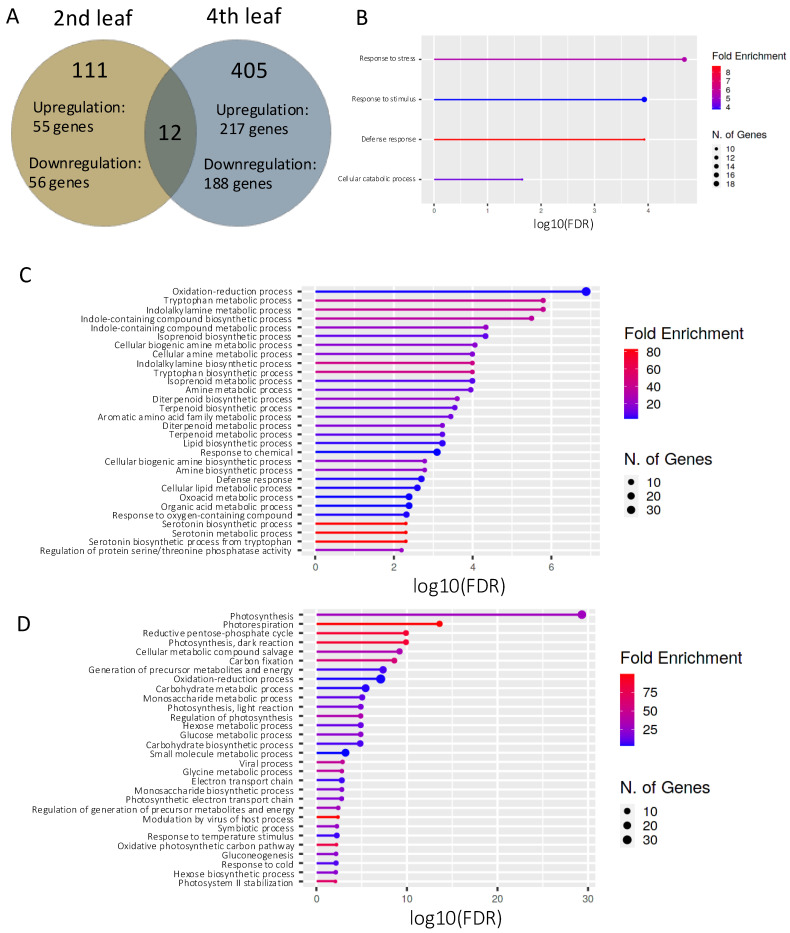
Transcriptome analysis of second and fourth leaves infected with blast fungus. (**A**) Venn diagram of differentially expressed genes (DEGs) in second and fourth leaves infected with the fungus. (**B**) Gene Ontology (GO) enrichment analysis of upregulated DEGs identified in second leaves. (**C**,**D**) GO enrichment analysis of upregulated DEGs (**C**) and downregulated DEGs (**D**) identified in fourth leaves. Circle size indicates the number of genes enriched in each GO term.

**Figure 3 plants-14-00241-f003:**
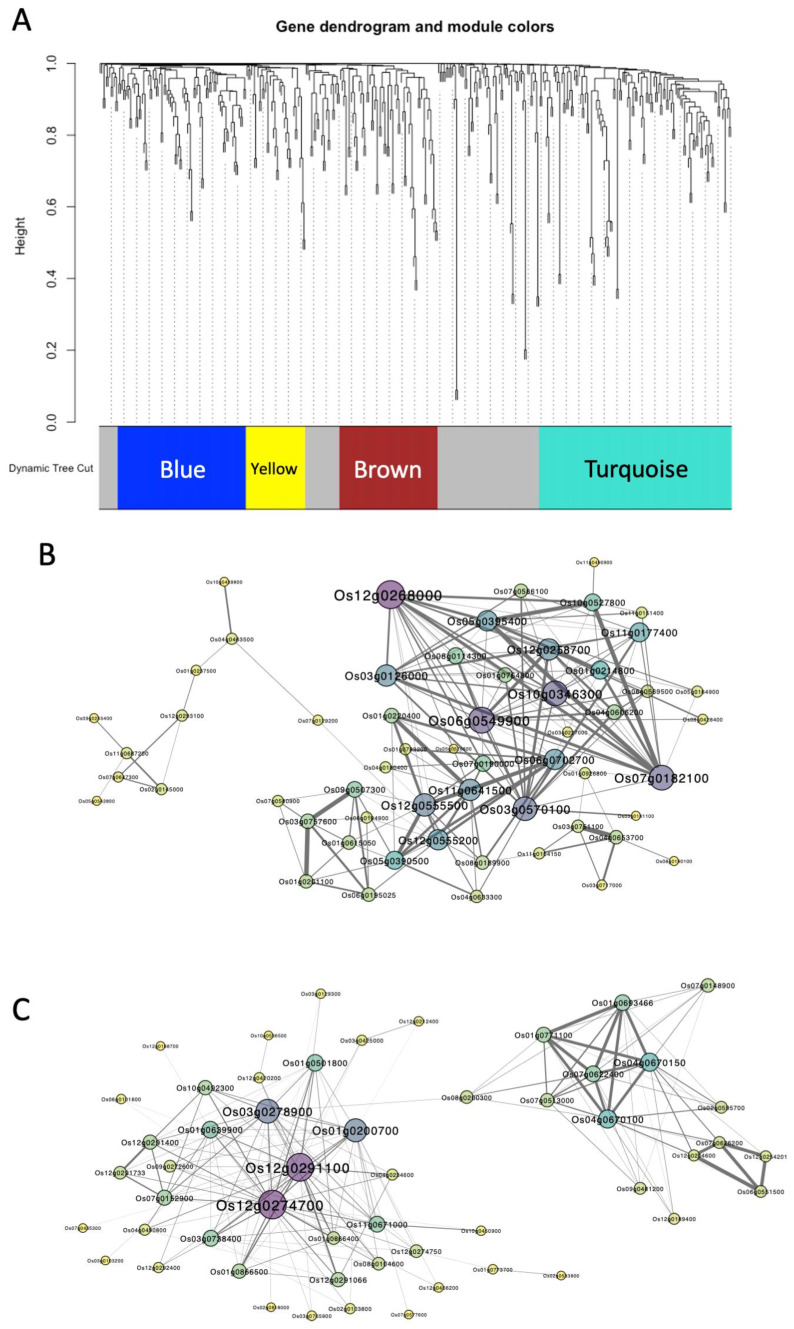
Weighted gene co-expression network analysis (WGCNA) of DEGs identified in fourth leaves infected with rice blast fungus. (**A**) Clustering dendrogram of 405 genes, with assigned module colors. (**B**,**C**) Networks of 57 genes in the blue module (**B**) and 53 genes in the turquoise module (**C**). Circle size indicates the connectivity of each node; line thickness indicates the weight of each edge.

**Table 1 plants-14-00241-t001:** List of DEGs common in the second and the fourth leaves. Normalized counts were obtained using the DESeq2 count function.

Gnen ID	log2 Fold Change		Annotation	Average Normalized Conunts			
				Second Leaf		Fourth Leaf	
	Second Leaf	Fourth Leaf		Control	Innoculated	Control	Innoculated
Os01g0137800	1.79	1.46	Non-protein coding transcript.	86.68	297.54	82.97	223.09
Os01g0615100	2.23	1.83	Chymotrypsin protease inhibitor, Salt and osmotic stress tolerance	247.76	1163.20	154.61	537.25
Os03g0661600	1.86	2.47	Similar to Alpha-amylase/trypsin inhibitor	65.10	235.33	55.41	299.47
Os04g0688100	1.49	1.08	Peroxidase	188.34	527.14	194.20	403.33
Os08g0135900	1.14	2.34	Similar to Tryptophan synthase beta chain 1	313.32	687.44	166.04	848.42
Os08g0136001	1.10	2.33	Hypothetical gene	303.20	649.83	164.72	833.49
Os10g0416500	1.48	2.39	Class IIIb chitinase	206.51	572.96	180.40	936.63
Os12g0437800	1.86	1.92	Protease inhibitor, MG1 (M. GRAMINICOLA-RESISTANCE GENE 1)-mediated nematode resistance	254.36	926.52	66.24	252.38
Os12g0448900	1.70	1.39	Fatty acid alpha-dioxygenase family, Enzyme that oxygenates fatty acids into 2R-hydroperoxides	145.91	469.06	135.98	353.71
Os12g0583300	8.11	8.28	Peptidase aspartic, catalytic domain containing protein.	0.00	55.85	0.00	43.18
Os07g0218200	−1.56	1.44	Terpene synthase, Disease resistance	232.29	78.42	71.46	192.03
Os02g0695600	−3.91	−2.71	Conserved hypothetical protein	106.96	7.40	156.33	24.01

## Data Availability

The RNA sequencing data are available in DDBJ under BioProject accession PRJDB19723.

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
