# Peer review of "Differential Stress Responses to Rice Blast Fungal Infection Associated with the Vegetative Growth Phase in Rice"

_plants, 2025, doi:10.3390/plants14020241_

Round 1
Reviewer 1 Report
Comments and Suggestions for Authors
Comments and suggestions are detailed in the file attached.

Author Response
Comments 1:
Authors have hypothesized that reduced miR156 expression in adult rice tissue could be a reason for higher resistance to blast fungus than in the juvenile phase and have performed the RNA-Seq analysis of rice leaves after infection with the blast fungus at the juvenile and adult phases to study the molecular changes associated with the two developmental phases.
The major concerns of the study is that the hypothesis made by the authors does not match the results or conclusion made in the manuscript. It has already been studied earlier in several studies that blast fungal infection is less severe in adult phase than in the juvenile phase of rice. Hence, the novelty of the study stands out if the authors highlight the differences in genes associated with the developmental stages of rice and the infection with the fungus. Authors have not mentioned anything about miR156 or WRKY45 in their results or manuscript and hence authors need to either revise their hypothesis or relate their results to their hypothesis in a clear and detailed manner. The study would have great impact if authors could have included a miR156 mutant rice line to link the role of this gene with varied fungal infection at different developmental stages of rice.
Response 1: Thank you very much for your very important comment. As Reviewer 1 pointed out, there are several previous studies on changes in disease resistance as the growth stage progresses regarding rice blast and bacterial leaf blight, and we now understand that it is nonsense to hypothesize that disease resistance will become stronger as growth progresses. Therefore, we focused on changes in host responses associated with growth, which previous studies have not yet obtained sufficient knowledge about, and we have redefined the questions that will make our experimental results acceptable to readers.
Comments 2: In the introduction section, authors could include additional examples of earlier research on the role of microRNA and rice blast studied at different growth stages.
Response 2: We newly introduced previous studies regarding age-related resistance (new reference number [18-23]) and the first genetic evidence linking miR156 to age-related resistance obtained from the maize mutant Corngrass1 (new reference number [24-25]) in the introduction.
Comments 3: Line 56-57: “symbiosis” needs to be replaced by “biosynthesis”
Response 3: We revised from “symbiosis” to “biosynthesis” as Reviewer 1 pointed out.
Comments 4: Control image in Fig 1A has to be included.
Response 4: We added the control image in Figure 1A as Reviewer 1 pointed out.
Comments 5: Authors need to mention clearly if second and third leaves were considered as juvenile and fourth and fifth as adult phases. May be a diagrammatic representation of the leaves considered here in the study would be appropriate.
Response 5: In this study, the definition of ‘juvenile phase’ and ‘adult phase’ followed Itoh et al. (2005). They defined that the first and second leaf stages is the juvenile phase, the third to fifth leaf stages are intermediate, and the later stage is the adult phase, and we introduced their definition in the introduction (new reference number [2]) and the Materials and Methods, too. Therefore, we revised our previous description of the fourth leaf from “adult leaf” to “more adult leaf” or “later leaf” compared with juvenile leaf.
Comments 6: As infection is almost always strongly correlated to the amount of pathogen present in the tissue so authors need to provide a strong reason of why increased fungal DNA was observed in adult leaves in spite of having less infection. Authors could quote some references of studies where such observation were made.
Response 6: As Reviewer 1 pointed out, we also observed the positive correlation between the lesion area and estimated fungal biomass in each growth stage, but this relationship clearly changed between the third and fourth leaf stage (please see new Figure 1D). To reveal the cause of this change, we observed the lesion area using microscopy and found that the fungal hyphae appeared to spread only on the leaf surface on second leaves. On the other hand, many conidia were observed in lesions on fourth leaves, whereas few conidia were found in lesions on second leaves. These results suggest that the status of the infecting fungus differs depending on the leaf age, which imply that plants at different stages exhibit different responses to rice blast infection. We consider that this result is a part of novelties in this study and critical to suggest the different status of blast fungus depending on the leaf age, so the result is newly added as Figure 1D. We speculate that the change in the relationship between the lesion area and the fungal biomass is greatly influenced by the density of conidia, which we also mentioned in the manuscript.
Comments 7: Line 101-102 has to be revised. Instead of using strong statements like “conclude” authors have use something like “we hypothesize that these differences are likely….”
Response 7: We revised from “conclude” to “hypothesize” as below;
2.1 Evaluation of rice blast resistance in leaves at different growth stages
We hypothesize that these differences are likely the result of complex physiological differences between juvenile and adult leaves,
Comments 8: Fig 1G-units of fungal mass has to be provided on Y-axis.
Response 8: In estimating the fungal biomass using qPCR, the amount of fungal DNA is not absolutely quantified, but fungal biomass is relatively quantified by comparing with that of the second leaf, so there are no units. For the easier understanding of readers, we revised the label of Y-axis from “Fungal mass” to “Relative fungal biomass”, and added more detailed explanation in Materials and Methods as below;
4.5 Disease severity assessment by qPCR
A standard curve was created through serial dilution of DNA extracted from the second leaf (Supplementary Figure 3I). The relative MHP1 gene DNA concentration of each sample to that of the second leaf was determined based on this standard curve and calculated crossing point values.
Comments 9: Line 129: log2 (fold change) has to be written as log2 fold change
Response 9: We revised from “log2 (fold change)” to “log2 fold change” as Reviewer 1 pointed out.
Comments 10: Table 1 is not clearly visible. Please provide a better one.
Response 10: We replaced Table 1 with higher resolution one as Reviewer 1 pointed out.
Comments 11: In the transcriptomic study, was the second leaf control compared to the second leaf inoculated one and is it the same way with the fourth leaf?
Response 11: Yes, the DEGs were detected by comparing inoculated second leaf with control second leaf, and also by comparing inoculated fourth leaf with control fourth leaf. For the easier understanding of readers, we revised our description in the Results as below;
2.2 Transcriptome dynamics in rice leaves at different stages under rice blast fungus infection
Differentially expressed genes (DEGs) were identified by comparing inoculated second leaves with control second leaves and by comparing inoculated fourth leaves with control fourth leaves based on the criteria of adjusted P < 0.05 and log2 fold change > 1.
Comments 12: In Section 2.2, in addition to describing the differential gene expression common in second and fourth leaves, it would be appropriate to mention the fold changes of these selected genes observed between these two growth stages and relate them to the disease incidence observed.
Response 12: Thank you very much for the valuable comment. As Review1 pointed out, the expression level of common 12 DEGs between the second and fourth leaf stage are likely different between the two leaf stages, so we performed DEG analysis between the control second leaf and control fourth leaf. However, these 12 DEGs did not show significant difference due to the higher adjusted P, so it seems difficult to mention the expression change of these 12 DEGs between leaf stages.
Comments 13: Authors need to provide the list of differentially expressed genes of control and inoculated samples, list of genes used for network analysis as a supplementary excel file.
Response 13: We newly prepared the list of 405 DEGs detected only in the fourth leaf, which we used for WGCNA, as Supplementary Table 2. In addition, we also prepared the list of 111 DEGs detected only in the second leaf as Supplementary Table 1.
Comments 14: Please follow the same nomenclature of the pathogen throughout the manuscript. In supplementary fig 3, the name of the pathogen used is different.
Response 14: We revised the nomenclature of the pathogen in Supplementary Figure 3 as Reviewer 1 pointed out.
Comments 15: Line 322: Please provide a reference for Smirnov–Grubbs tests used to measure the lesion area
Response 15: We added the reference for Smirnov-Grubbs tests (new reference number [45]) as Reviewer 1 pointed out.
Comments 16: Please mention what was the housekeeping gene used in the qPCR studies and also provide a reference for the protocol used for fungal DNA isolation.
Response 16: The fungal biomass was estimated from the concentration of MHP1 DNA contained in the total DNA extracted from infected rice leaves. Since the DNA of the rice blast fungus is extremely small compared to that of the host, housekeeping genes were not used to compensate for the amount of DNA on the host side. As mentioned in the Materials and Methods, fungal DNA was extracted with rice DNA from infected leaves and was not isolated from rice DNA. Nevertheless, we added the reference for the quantification of MHP1 gene in the Materials and Methods (new reference number [29]).
Comments 17: Was the whole leaf with infection used for the qPCR analysis or a section of leaf with the lesion was used?
Response 17: We used whole leaf blades containing the lesion area for DNA extraction. We revised the description of this point in the Materials and Methods as below;
4.5 Disease severity assessment by qPCR
The entire leaf blades, including all lesions, were sampled 6 days after inoculation;
Comments 18: In the methods section, authors need to provide details about the number of biological and technical replicates and how many plants per experiment were used for each experiment.
Response 18: The number of biological replicates for disease severity assessment was different depending on the leaf stage, so the detailed number was already described in the legend of Figure 1. On the other hand, we analyzed three technical replicates for disease severity assessment by qPCR and three biological replicates for RNA-seq analysis, which were not described in the previous manuscript. So, we newly added in the Materials and Methods as below;
4.5 Disease severity assessment by qPCR
Three technical replicates were analyzed for each experimental group.
4.8 RNA-seq analysis
Three biological replicates were analyzed for each experimental group.
Comments 19: Authors need to submit the RNA-Seq data to NCBI and obtain the archive ID.
Response 19: We already submitted our RNA-seq data to DDBJ and mentioned in the section of Data Availability Statement as below;
The RNA sequencing data is available in DDBJ under BioProject accession PRJDB19723.
Reviewer 2 Report
Comments and Suggestions for Authors
The manuscript entitled " Differential Stress Responses to Rice Blast Fungal Infection Associated with the Vegetative Growth Phase in Rice"
Authors have well executed the experiment plan of blast inoculation, measuring the fungal biomass, trypan blue staining, etc which is very interesting. In my opinion, this manuscript has a good potential for publishing. I have few minor questions to the authors.
Lines15-17: authors mentioned ‘Enhanced resistance to rice blast fungal infection following miR156 15 knockdown in rice has been attributed to upregulation of WRKY45 downstream of miR156. There- 16 fore, it was hypothesized that miR156 downregulation in the adult phase would result in higher rice 17 blast resistance than in the juvenile phase’.
I can understand, based on the earlier results of mir156 authors took this hypothesis to study in this manuscript. But in the abstract section authors should summarize the present paper method and result. Authors have not conducted any experiment to validate the mir156 and WRKY45 gene in the present study. So please remove these sentences from the abstract.
Line121: Table 1 is not clear. Please add the clear table in the revised manuscript.
Line 283-287: Authors mentioned leaves turn yellow when it reaches six leaves stage. Mention the number of days when the plant reached six leaves stage. From the single plant you have collected the 2,3,4,5th leaves? Is it due to long duration the plants turned to yellow in the growth camber? Explain more clearly in the methods.
Lines 362-366: Author mentioned about the incompatible and compatible interaction. It leads to a misunderstanding of collecting both resistant and susceptible varieties for RNA sequencing. Please remove it. How many biological replications was used for RNA sequencing?
Line 344: In the side heading ‘Trypan blue stainin’. Please check the spelling ‘g’ is missing. Correct in the revised manuscript.
Author Response
Comments 1:
The manuscript entitled “Differential Stress Responses to Rice Blast Fungal Infection Associated with the Vegetative Growth Phase in Rice”
Authors have well executed the experiment plan of blast inoculation, measuring the fungal biomass, trypan blue staining, etc. which is very interesting. In my opinion, this manuscript has a good potential for publishing. I have few minor questions to the authors.
Lines15-17: authors mentioned ‘Enhanced resistance to rice blast fungal infection following miR156 15 knockdown in rice has been attributed to upregulation of WRKY45 downstream of miR156. There- 16 fore, it was hypothesized that miR156 downregulation in the adult phase would result in higher rice 17 blast resistance than in the juvenile phase’.
I can understand, based on the earlier results of mir156 authors took this hypothesis to study in this manuscript. But in the abstract section authors should summarize the present paper method and result. Authors have not conducted any experiment to validate the mir156 and WRKY45 gene in the present study. So please remove these sentences from the abstract.
Response 1: Thank you very much for your very important comment. In the previous manuscript, we constructed the logic based on the hypothesis that rice blast resistance may change as the growth phase progress. But after that, we found several previous studies on changes in disease resistance as the growth stage progresses regarding rice blast and bacterial leaf blight, and we now understand that it is nonsense to hypothesize that disease resistance will become stronger as growth progresses. Therefore, we focused on changes in host responses associated with growth, which previous studies have not yet obtained sufficient knowledge about, and we have redefined the questions that will make our experimental results acceptable to readers. Along with that, the descriptions of miR156 or WRKY45 were removed from the Abstract.
Comments 2: Line121: Table 1 is not clear. Please add the clear table in the revised manuscript.
Response 2: We replaced Table 1 with higher resolution one as Reviewer 2 pointed out.
Comments 3: Line 283-287: Authors mentioned leaves turn yellow when it reaches six leaves stage. Mention the number of days when the plant reached six leaves stage. From the single plant you have collected the 2,3,4,5th leaves? Is it due to long duration the plants turned to yellow in the growth camber? Explain more clearly in the methods.
Response 3: Under our growth condition, the color of sixth leaf is not as good as those of the previous leaves probably due to the long duration (three weeks), so we used from the second to fifth leaves for the experiments. In addition, each leaf is derived from the separate individuals when the plants reach each leaf stage, which was not described in the previous manuscript. So, we revised the Materials and Methods as below;
4.1 Rice growth conditions
Under our growth conditions, leaves turned yellow when the rice plants reached the sixth leaf stage due to the long duration (three weeks) in the growth chamber; therefore, rice plants at the second to fifth leaf stages were used. The latest fully developed leaves at each leaf stage from separate individuals were used for the experiments.
Comments 4: Lines 362-366: Author mentioned about the incompatible and compatible interaction. It leads to a misunderstanding of collecting both resistant and susceptible varieties for RNA sequencing. Please remove it. How many biological replications was used for RNA sequencing?
Response 4: The confusing description for incompatible and compatible interaction was removed from the Materials and Methods as Reviewer 2 pointed out. For the RNA sequencing, three biological replicates were analyzed for each experimental group, which was not described in the previous manuscript. So, we newly added in the Materials and Methods as below;
4.8 RNA-seq analysis
Three biological replicates were analyzed for each experimental group.
Comments 5: Line 344: In the side heading ‘Trypan blue stainin’. Please check the spelling ‘g’ is missing. Correct in the revised manuscript.
Response 5: We revised the misspelling as the Reviewer 2 pointed out.
Round 2
Reviewer 1 Report
Comments and Suggestions for Authors
Authors have addressed most of the concerns and suggestions recommended earlier. New comments are included in the document attached.

Author Response
Comments 1: Authors have addressed most of the comments/suggestions and have made the necessary corrections. However, it is still not convincing how there could be less fungal DNA from highly infected samples. Authors claim that there were more hyphae in juvenile leaves and there were more conidia in intermediate leaves, but authors need to understand that hyphae contain more DNA compared to conidia (which are haploids) and the explanation does not stand well for the observations made. Hence their results in Section 2.1 needs to be revised with appropriate explanation.
Response 1: Thank you very much for your very important comments. Since this result is a very important part of this paper, we need to explain in more logical and acceptable way to readers. When we looked back at the results, we realized that the reason hyphae were so conspicuous on the second and third leaves was mainly due to the low density of the hyphae themselves. Furthermore, the difference in hyphal density also explains why the lesion area and estimated fungal biomass do not match. So, we replaced Figure 1E-H with Figure 1E and F, which show the difference in hyphal density between the second and fifth leaves more clearly, and revised the description as below;
2.1 Evaluation of rice blast resistance in leaves at different growth stages
Compared with the second leaf, the hyphae with conidia were more densely spread around the inoculated spot in the fifth leaf (Figure 1E and F). Therefore, it was assumed that the difference in hyphal density was the cause of the discrepancy between the lesion area and fungal biomass.
- Discussion
This discrepancy can be explained by the more fungal biomass per lesion area in older plants than in juvenile plants due to the higher hyphal density (Figure 1D-F).
Comments 2: Line 16-17: Modify as “While much knowledge has been obtained about changes in disease resistance as growth stages progress, knowledge about changes in plant responses to pathogens with progressing plant’s age is limited”.
Response 2: We revised this sentence as Reviewer 1 suggested.
Comments 3: Line 22-23: Modify as “These findings highlight differences in rice leaf stress responses to blast infection at different growth stages”.
Response 3: We revised this sentence as Reviewer 1 suggested.
Comments 4: Line 59-60 and 76-77 are repetitive as the abstract so please rephrase it.
Response 4: We rephrased the sentences in the Introduction as below;
Line 59-62
Generally, plants are more susceptible to disease in early phase than in late phase, and resistance to disease increases during the process of maturation [18-20]. This increase could be due to the gain or reinforce the ability to control the infection and/or proliferation of pathogen, which is known as adult plant resistance or age-related resistance [21-23].
Line 77-79
Although much knowledge has been obtained regarding the changes in disease resistance through the maturation of plants, there is limited understanding of changes in plant responses to pathogens with progressing plant’s age.
Comments 5: In line 78-79: please remove “during the vegetative phase”
Response 5: We revised this sentence as Reviewer 1 suggested.
Comments 6: Instead of writing as “later stages” please address as “intermediate stage” as fourth and fifth leaves were used in this study
Response 6: We replaced the description of “later stages” with “intermediate stage” as Reviewer 1 suggested.
Comments 7: In line 90: please remove “at each leaf stage” because the leaf number mentioned here itself describes the stage of the plant
Response 7: We revised this sentence as Reviewer 1 suggested.
Comments 8: Line 95: modify “do more adult leaves” as “the older leaves”
Response 8: We revised this sentence as Reviewer 1 suggested. In addition, we replaced the description of “more adult leaves” in other parts with “older leaves”.
Comments 9: Figure 1 title could be modified as “P. oryzae infection in different growth stages of rice”
Response 9: We revised the title as Reviewer 1 suggested.
Comments 10: Figure 1A legend could be improved as “Lesions on leaves at different plant growth stages. The control leaves were treated with sterilized water. Scale bar = 10 mm”.
Response 10: We revised the legend as Reviewer 1 suggested.
Comments 11: Fig 2A legend could be improved as “Venn diagram of differentially expressed genes (DEGs) in second and fourth leaves infected with the fungus”
Response 11: We revised the legend as Reviewer 1 suggested.
Comments 12: Authors have provided the list of DEGs from second and fourth leaves. It is advised to mark the genes in supplementary table 2 with different modules as well so that it is convenient for the readers to understand what are the genes used for the network analysis.
Response 12: Thank you for your suggestion. We added the column of “Module Colors” in Supplementary Table2.
Comments 13: The Bioproject ID PRJDB19723 could not be verified so please provide the correct information.
Response 13: The sequence data has already been registered with that number, but it was kept confidential until the manuscript was accepted. Since we think that they may be necessary for reviewing, we released the data. Please find “PRJDB19723” on NCBI, etc.